# Towards an Understanding of Physical vs Virtual Robot Appendage Design

Zhao Han*, Albert Phan*, Amia Castro*, Fernando Sandoval Garza* and Tom Williams

*MIRRORLab, Department of Computer Science, Colorado School of Mines*, Golden, Colorado, USA 80401

zhaohan@mines.edu, aphan@mines.edu, acastro@mines.edu, fsandovalgarza@mines.edu, twilliams@mines.edu

*Abstract*—Augmented Reality (AR) or Mixed Reality (MR) enables innovative interactions by overlaying virtual imagery over the physical world. For roboticists, this creates new opportunities to apply proven non-verbal interaction patterns, like gesture, to physically-limited robots. However, a wealth of HRI research has demonstrated that there are real benefits to physical embodiment (compared, e.g., to virtual robots displayed on screens). This suggests that AR augmentation of virtual robot parts could lead to similar challenges.

In this work, we present the design of an experiment to objectively and subjectively compare the use of AR and physical arms for deictic gesture, in AR and physical task environments. Our future results will inform robot designers choosing between the use of physical and virtual arms, and provide new nuanced understanding of the use of mixed-reality technologies in HRI contexts.

*Index Terms*—augmented reality (AR), mixed reality (MR), deictic gesture, non-verbal communication, physical embodiment, robotics, mobile robots, human-robot interaction (HRI)

## I. INTRODUCTION

To gain trust and acceptance, robots must be able to effectively communicate with people. Due to robots' unique physical embodiment [1], human-robot interaction (HRI) researchers have investigated non-verbal behaviors [2], such as implicit arm movement (e.g., [3], [4]), gestures [5], and eye gaze [6], [7]. Multimodal approaches pairing these nonverbal displays with verbal communication have also been well-studied (e.g., [8]–[10]). Results show that non-verbal behaviors themselves are particularly important as they increase task efficiency [6] and improve subjective perceptions of robots [3].

Unfortunately, most robot systems – such as mobile or telepresence robots, autonomous vehicles, or free-flying drones – do not have the physical morphology to express these types of nonverbal cues, lacking heads and eyes for gazing, or arms for gesturing. Moreover, the high degree-of-freedom requirements and complex mechanics of these morphological components, especially physical arms, present cost barriers, especially when such components would only be used for gesturing and not for manipulation. Finally, inclusion of physical components like arms presents well-known safety concerns [11].

To address these challenges, researchers have investigated *virtual* counterparts. For nonverbal facial cues, this has taken a variety of forms. The Furhat robot head [12], for example, uses projection mapping to display a humanlike face without

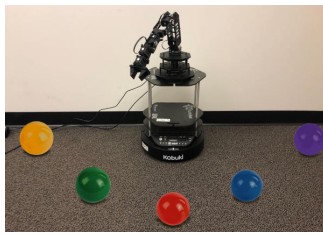

(a) Physical Robot with a *physical arm* pointing to a *physical referent* (P→P)

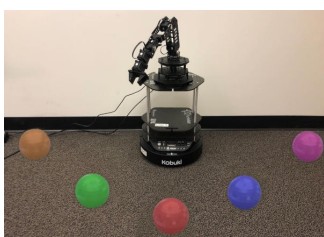

(b) Physical Robot with a *physical arm* pointing to an *AR virtual referent* (P→AR)

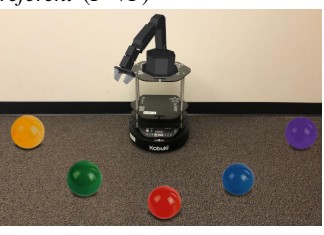

(c) Physical Robot with a *AR virtual arm* pointing to a *physical referent* (AR→P)

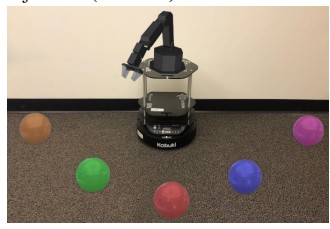

(d) Physical Robot with a *AR virtual arm* pointing to an *AR virtual referent* (AR→AR)

Fig. 1: Artist's rendering of the four conditions to be investigated. To investigate the intersection of physical and AR worlds, with a focus on referring behavior (physical/AR virtual *arm* × physical/AR virtual *referent*), we present an experiment design to evaluate such four interactions at the crossing of the real and mixed worlds. Note that the virtual model will be made hollow, like the physical real arm.

requiring actuation. Similarly, many approaches use tablets to display a robot's face (e.g., [13], [14]). Finally, some researchers have leveraged augmented reality to display and customize robots' faces [15].

Augmented Reality (AR) has also been recently used to provide a lower-cost alternative for enabling gestural capabilities. For example, Groechel et al. [11] studied the use of an AR arm on a mobile robot, and Hamilton et al. [16] further considered the use of AR arms for deictic gesture, comparing AR arms to other types of AR annotations (e.g., arrows [9]). Results showed that arms were better perceived subjectively. Yet, the performance differences between virtual (AR) and physical arms has not yet been explored. This means that while the monetary cost differences between these options can be readily compared, the performance differences between these platforms is not yet well understood, presenting a challenge for robot developers.

*The first four authors contributed equally. This work has been supported in part by the National Science Foundation under IIS-1909864.

Moreover, it is further unclear whether differences between virtual and physical arms might be contingent on the virtuality or physicality of the task-relevant objects to which a robot might choose to gesture. It could be the case, for example, that virtual arms might be viewed more positively when used in tasks involving virtual referents, and vice versa. This would be a potentially complex challenge to reason over given that mixed-reality task environments may contain a mixture of virtual and physical objects.

In this paper, we present a study design to investigate the differences in objective performance and subjective perception between physical and virtual (AR) arms, as mediated by the physicality or virtuality of the robot's target referent (See Figure 1). This work will help robot designers to better understand whether and when to employ virtual rather than physical morphological components. Moreover, this will help provide more precise design guidelines that are sensitive to the nuances of mixed-reality robotics environments.

## II. RELATED WORK

### A. Virtual vs. Physical Agents

Much HRI research has already demonstrated differences in objective performance and subjective perception between purely virtual and purely physical robotic entities. Much of this work has compared physically embodied robots to virtual agents depicted on screens. This research demonstrates that embodied physical presence leads to greater influence [17], learning outcomes [18], task performance [19], [20], gaze following from infants [21], proximity [17], exercise [22], pervasiveness [23], positive perception [23], and social facilitation [24], forgiveness [24], enjoyableness [1], [22], [25], helpfulness [22], [26], and social attractiveness [22]. However, these works have not considered morphologies that blend the physical and the virtual, as is enabled by AR Technologies.

### B. Virtual Agents in Augmented Reality

Different from the virtual agents wholly residing inside the virtual world, AR allows virtual objects/agents to be projected onto a user's view of the real world [27]. A variety of research has examined how interactants perceive agents in AR. Obaid et al. [28] showed that AR agent is perceived as physically distant where participants adjust their sound level accordingly, and Kim et al. [29] showed that AR agents aware of physical space were rated higher in social presence.

Other researchers have examined how people perceive virtual humans (ostensibly) interacting with the physical world through AR. Lee et al. [30] studied AR-visualized humans subtly moving a physical table in terms of presence, co-presence, and attentional allocation. Schmidt et al. [31] experimented with virtual humans manipulating physical objects: hitting a physical ball with a virtual golf club, but did not find statistical significance in realism and emotional responses. In contrast, our work considers a physical entity (a physical robot) with a virtual appendage, rather than a wholly virtual agent.

### C. Augmented Reality for Robot Communication

In this work we are moreover specifically interested in robots using AR appendages for the purpose of communication. There has been a variety of work on AR for robot communication within the broader area of VAM-HRI [32], [33]. Frank et al. [34] used AR to show reachable spatial regions, to allow for humans to know where and when to pass objects to robots. Taylor et al. [35] used AR to remove robot arm occlusion and see occluded objects by making the arm transparent. Diehl et al. [36] used AR to verify learned behavior in the robot learning domain to increase safety and trust. In addition to AR using headsets, researchers have investigated projected AR. For example, [37] used projected AR to project car door frame, moving instructions, and task success in a car-assembly collaborative application. And Han et al. has focused on open science, making projector-based AR more readily available [38], [39].

### D. Human and Robot Deictic Gesture

Finally, within this broader area, our work specifically examines robots' use of AR visualizations for the purposes of deictic gesture. Deictic gesture has been a topic of sustained and intense study both in human-human interaction [40], [41] and human-robot interaction [5]. Deictic gesture is a key and natural communication modality, with humans starting to use deictic gesture around 9-12 months [42], and mastering it around age 4 [43]. Adults continue to use deixis to direct interlocutor attention, so as to establish joint and shared attention [44]. As a non-verbal modality, gesturing is especially helpful in public noisy environments such as factories, warehouses, or malls [2], [45]. Accordingly, roboticists have leveraged this and study its effects for better understandable robots, e.g., in tabletop environments [46] and free-form direction-giving [47]. Specifically, research shows that robot, like humans, can shift interlocutor attention [48] and can use a variety of deictic gestures, not only pointing [5], [49]. Williams et al. have begun to explore the use of deictic gesture within Augmented Reality [9], [50]–[53], although most of this work has been with non-anthroporphic visualizations like virtual arrows. In contrast, Hamilton et al. [16], like ourselves in this work, recently examined virtual arms, and showed that AR virtual arm enhanced social presence, likability. Unlike Hamilton, however, we are interested in explicitly comparing virtual arms to physical arms (rather than other types of virtual gestures) and in understanding the role that the physical or virtual nature of the environment might mediate these differences.

## III. HYPOTHESES

We approach this work with a set of key hypotheses and expectations. First, we maintain hypotheses regarding the objective effectiveness of AR gestures.

**Hypothesis 1 (H1) – Equal Accuracy.** We believe that robot deixis with a virtual arm will be no less accurate than deixis with a physical arm.

**Hypothesis 2 (H2) – Reality Alignment mediates Efficiency.** We believe that while using physical or virtual arms to

refer to physical or virtual referents, *respectively*, should have equivalent efficiency, we hypothesize that a mismatch between these levels of reality (i.e., virtual arms pointing to physical objects, and vice versa) could increase the time needed by users to identify the robot's target, due to a need for additional cognitive processing to explicitly overcome this misalignment.

**Hypothesis 3 (H3) – Reality Mediates Perception.** Similarly, we believe a mismatch between gesture-reality and referent-reality will reduce perceived naturality and likability.

## IV. METHOD

To investigate these hypotheses, we plan to employ the following experimental method.

### A. Apparatus

*1) Robot Platform:* As one of the application domains of this work is a gesturally-limited robot, we plan to use a TurtleBot 2 mobile robot [54]. This differential wheeled robots is the second generation of the Turbot family, and is maintained by the current maintainer of Robot Operating System (ROS) [55], thus having a large support community. The specification for TurtleBot compatible platforms can be found on ros.org [56].

*2) Mixed-Reality Head-Mounted Display (MR-HMD):* Microsoft HoloLens 2 [57] will be used as our MR-HMD. It is a commercial-grade see-through holographic mixed reality headset with a $43° \times 29°$ Field of View (FOV).

*3) Physical/Virtual Robot Arm:* The physical arm we will use will be the WidowX Robot Arm [58]: a 5-DoF arm with a parallel gripper, which an reach up to $41cm$ horizontally and $44cm$ vertically. Our virtual arm will be created using the CAD models and Unified Robot Description Format (URDF) model of this arm [59], as rendered in Unity. Although the WidowX is a relatively simple arm[1], it is relatively costly as a deictic appendage, priced at $1,699.95 USD.

The virtual arm will have the same distance to the TurtleBot top when rendered in Unity. To affix the AR virtual arm to the same position as the physical arm, we plan to place a $12cm$ cardboard cube on the top panel of the TurtleBot2. Each face of the cube has fiducial markers for localization [60].

*4) Physical/Virtual Referents:* Five spheres [61] will be used as communicative referents and will be arced within the field of view of HoloLens 2 (See Figure 1). Each sphere measures $d = 15.24cm$ ($6in$) in diameter and will be placed $45°$ apart, as shown in Fig 1. The distances between the robot and the referents are preliminarily determined to be $3 \times d$ for clarity and $1m$ within field of view. The physical and rendered spheres will have the same size and placement.

### B. Gesturing Task and Implementation

These materials will be used in the context of a standard gesture-comprehension experiment. In each trial, the WidowX robot arm, mounted or simulated on top of TurtleBot 2, will randomly point to one of the colored spherical targets, which

---

[1]See other arms on https://www.trossenrobotics.com/robotic-arms.aspx for a comparison of prices.

TABLE I: Four Study Conditions across Two Dimensions on Physicality/Virtuality

|  |  | **Referent Virtuality** | |
|---|---|---|---|
|  |  | *Physical* | *AR Virtual* |
| **Arm Virtuality** | *Physical* | P→P | P→AR |
|  | *AR Virtual* | AR→P | AR→AR |

participants will then be asked to identify by air-tapping on that target. This will be repeated ten times, with targets chosen at random. While a controller could be used, the four directional buttons do not work well for five referents, and would introduce confounds for measuring response time.

For each gesture, the MoveIt motion planning framework [62] will be used to move the end effector to the desired pointing pose. As we plan to conduct this experiment in person, the trajectory generated by MoveIt to its final pose, non-deterministic due to the probabilistic algorithms [63], will be made deterministic by specifying multiple waypoints before the pointing poses towards different spheres. This approach to a deterministic outcome has seen success in robot-to-human handover tasks [14]. As an alternative approach, we are also investigating recording the trajectory output by MoveIt given both the placement of the robot and the spheres are static, thus the recorded trajectory will be replayed at experiment time.

For the AR virtual arm, i.e., the WidowX arm model rendered in Unity, we plan to implement the functionality that converts the MoveIt trajectory to key-frame-based animation either by subscribing to MoveIt output trajectory directly or from a pre-recorded trajectory.

The MoveIt and Unity implementation will be open-sourced on GitHub to facilitate replication.

### C. Experiment Design

This study will follow a $2 \times 2$ *between-subjects* design because a within-subjects design would require rapidly uninstalling the physical arm and cardboard cube from the top panel of the robot, which would take significant effort and time, and could be error-prone within the short timeframe of an experiment.

As implied throughout this work so far, we will manipulate whether the arm and the referent, i.e., the spheres, are physical or rendered. Formally, two independent variables will be manipulated: *referee virtuality* and *referent virtuality*. Thus, there will be four study conditions across the two factors:

- P→P (Physical): *Real* arm pointing at *real* spheres
- P→AR: *Real* arm pointing at *virtual* spheres
- AR→P': *Virtual* AR arm pointing at *real* spheres
- AR→AR: *Virtual* AR arm pointing at *virtual* spheres

### D. Procedure

The study will be conducted in person in order to use the head-mounted display. All apparatus will be disinfected before use due to COVID-19 concerns.

Upon arrival, participants will be presented with informed consent information, in which they will be asked to identify the object the robot is referring to quickly and accurately to

impose the same amount of time pressure. After agreeing to participate, they will fill out a demographic survey and be randomly assigned to one of the four experimental conditions.

After watching a video on how to put on HoloLens 2, they will wear the headset and receive further training. The experimenter will first briefly get participants familiar with the task again as described in the informed consent form. Then participants will experience an interactive tutorial on HoloLens 2, designed in Unity. It will 1) allow participants walk through sample experiment trials to see how either the physical or virtual arm moves to gesture, 2) allow eye-tracking user calibration to accurately collect accuracy data (See Section IV-E1 below), and 3) get participants familiar with the air-tap gesturing to confirm the target object they believe in. While this is an onboarding experience, it is also considered to avoid novel effects. Experimenters will ask clarifying questions to ensure their understanding of the task and the procedure. After completing 10 trials, they will be asked to answer a questionnaire with all the subjective measures. At the end, participants will be paid according to our planned pilot study duration and debriefed.

### E. Data Collection and Measures

To test our hypotheses, we plan to collect two objective metrics and five subjective metrics to capture experience. Some of the subjective metrics are inspired by [16].

*1) Accuracy:* To facilitate data gathering for accuracy to measure effectiveness, participants in every condition will wear an HoloLens 2 in order to assess whether they have inferred which referent the robot has pointed at. This is achieved by the built-in eye-tracking API [64] of HoloLens 2. To collect such gazing information, the same numbers of invisible objects as the referents' will be implemented in Unity to be at the same positions as the rendered balls.

Accuracy will be calculated as the percentage of true positives where participants looked at the target referent in all trials, confirmed by "clicking" (using air-tap gesture) on the referent in their belief. The gesture will be detected by HoloLens 2's built-in gesture recognition capabilities.

*2) Reaction Time:* Similar to how accuracy is measured, reaction time will be measured through the HoloLens 2 eye-tracking API. Specifically, reaction time will be the duration between when the robot arm starts moving from its home position to when participants look at the target object.

*3) Social Presence:* As seen in Section II, social presence, defined as the feeling of being in the company of another social actor [65], has been a central metric in studies involving virtual agents. It can enable more effective social and group interactions [66], [67]. Within HRI, it has been found to increase enjoyment and desire to re-interact [68].

*4) Anthropomorphism:* Projecting human characteristics to non-human entities [69]–[71], such as attaching the AR virtual arm to the TurtleBot 2 in this work, encourage humans to re-use the familiar interaction patterns from human-human interactions. It facilitates sensemaking and mental model alignment [71], leading humans to be more willing to interact, accept, and

understand robot's behaviors [72]. Robots that use gesture have been found to appear more anthropomorphic [73]. Hamilton et al. [16] suggested that the robot with a virtual arm may have been viewed as more anthropomorphic, however it is unclear how this is compared with a robot with a physical arm. Anthropomorphism will be measured using the Godspeed Anthropomorphism scale [74].

*5) Likability:* As one of the primary metrics used in nonverbal robot communication [51], [73], [75], Likability summarize peoples' overall perceptions of technology, key to estimate people's experience. Similarly, Hamilton et al. [16] found evidence that the robot with a virtual arm enhanced likability, but it did not compare with the physical counterpart. Likability will be measured using the Godspeed Likability scale [74].

*6) Warmth and Competence:* As psychological constructs at the core of social judgment, warmth and competence are responsible for social perceptions among humans [76]. Warmth captures whether an actor is sociable and well-intentioned, and competence captures whether they can deliver on those intentions. Warmth and competence are thus key predictors of effective and preferable interactions, both for human-human interaction [76] and human-robot interaction [77], [78]. Moreover, they have been connected to social presence [79], and anthropomorphism [80], [81]. Warmth and Competence will be measured using the ROSAS Scale [82].

### F. Data Analysis and Participants

We plan to analyze the data within a Bayesian analysis framework [83] using the JASP 1.6 (version at submission time, will update) software package [84], with the default settings justified by Wagenmakers et al. [85]. Bayesian analysis with Bayes factors has benefits over the more common frequentist approach [83]. Some of them include that the Bayes factors can quantify evidence *for* the null hypothesis $\mathcal{H}_0$, and evidence *for* $\mathcal{H}_1$ vs. $\mathcal{H}_0$. In contrast, the $p$ value cannot provide a measure of evidence in favor of $\mathcal{H}_0$. For more details, we refer readers to [83]. All experimental data and analysis scripts will be made available publicly to facilitate replication.

Unlike the frequentist approach in need of power analysis to achieve sufficient power [86], [87], the Bayesian analysis does not strictly require power analyses to determine sample size [88] as it is not dependent on the central limit theorem. Nonetheless, we plan to recruit at least 25 participants, similar to the AR virtual arm vs. the AR virtual arrow study [16].

### V. CONCLUSION

In this workshop paper, we have described the design of an experiment designed to investigate the differences in performance between physically-limited robots that use either physical or virtual (AR) arms, as mediated by the physical or virtual nature of their target referents. Our immediate future work will be to conduct, analyze, and report the results of this experiment. Our hope is that our results provide new insights into mixed-reality human-robot interaction, and will help inform robot designers' decisions as to whether to use virtual or physical robotic arms.

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
