# OpenReview forum: "Towards an Understanding of Physical vs Virtual Robot Appendage Design"
_humanrobotinteraction.org/HRI/2022/Workshop/VAM-HRI — VAM-HRI 2022_

### Official Review · Reviewer_C7Zz · 2022-02-25
**Interesting and relevant proposed study, accept**

**Rating:** 8
**Confidence:** 5

**Review:**

The proposed study for evaluating the impact of virtual vs. real robot appendages for referring to virtual or real world objects is interesting and relevant to the VAM-HRI community. Overall, the paper is very well written and clear, and the study is well thought out, making this a clear accept.

Feedback:
1. In the abstract, the first mention of Augmented Reality and Mixed Reality should introduce AR/VR acronym
2. For the study, the alternative approach of recording the trajectory from MoveIt and replaying it seems more consistent and appropriate than replanning each time.
3. Why do the users use an air-tap to select the item? Why not let them press a button on a controller to indicate the object?
4. It would be worthwhile to motivate why visualizing a virtual arm for pointing is better than just highlighting the intended object (in the setting with a virtual arm), or devisualizing all other visualized objects in the scene besides the intended object (in the setting with virtual objects)

---

### Official Review · Reviewer_x3Si · 2022-02-28
**Interesting study builds on significant prior work, accept**

**Rating:** 8
**Confidence:** 5

**Review:**

This paper provides details on a proposed human subjects study. The researchers will compare a virtual robot arm to a physical one, and virtual referents to physical ones, in a 2 x 2 between subjects protocol. Measures collected include accuracy, reaction time, social presence, anthropomorphism, likability, and warmth & competence. This study has a number of implications, and the results will be interesting to see. Some questions follow that will hopefully assist the authors.

- Will participants know that their reaction time and accuracy are being assessed? How do you think that might change the outcomes?
- How will you decide the physical spacing of the referents to ensure clarity?

---

### Decision · Program_Chairs · 2022-03-04

Accept